

# Efficacy and safety of Brolucizumab for neovascular age-related macular degeneration: a systematic review and meta-analysis

Ran Dou and Jian Jiang

Department of Ophthalmology, Qilu Hospital (Qingdao), Cheeloo College of Medicine, Shandong University, Qingdao, China

## ABSTRACT

**Background:** To evaluate the efficacy and safety of Brolucizumab for neovascular age-related macular degeneration (n-AMD) through a systematic review and meta-analysis.

**Materials and Methods:** Cochrane, PubMed, Embase, and Web of Science databases were comprehensively searched for relevant studies. Stata and RevMan5.4 were applied for meta-analysis and risk of bias assessment. Data on the best-corrected visual acuity (BCVA), central subfield thickness (CSFT), presence of intraretinal fluid (IRF) and/or subretinal fluid (SRF), participants with ≥1 serious adverse events, and participants with ≥1 adverse events were analyzed.

**Results:** Six studies were finally included. Meta-analysis showed statistical differences in BCVA [SMD = −0.65, 95% CI [−0.17 to −0.23], $P < 0.05$], the presence of IRF and/or SRF [RR = 0.67, 95% CI [0.56–0.79], $P < 0.05$], and the safety of participants with ≥1 serious adverse events [RR = 0.57, 95% CI [0.39–0.84], $P < 0.05$] between the experimental group and the control group. However, no statistical differences were observed in CSFT [SMD = −1.16, 95% CI [−2.79 to 0.47], $P > 0.05$] or the safety of participants with ≥1 adverse events [RR = 1.07, 95% CI [0.97–1.17], $P > 0.05$].

**Conclusions:** Compared to other anti-VEGF drugs such as Aflibercept and Ranibizumab, intravitreal injection of 6 mg Brolucizumab is more effective and safer for n-AMD, especially in the presence of IRF and/or SRF, and for participants with ≥1 serious adverse events.

Corresponding author
Jian Jiang, thisisjiangjian@126.com

## INTRODUCTION

Neovascular age-related macular degeneration (n-AMD) is widely recognized as a leading cause of blindness (*Sharma et al., 2021*; *Takahashi et al., 2022*; *Ricci et al., 2020*). Over the last decade, the first-line therapy for n-AMD recommended by the treatment guidelines is the intravitreal injection of anti-vascular endothelial growth factor (VEGF) agents (*Tamashiro et al., 2022*; *Wong et al., 2008*), including ranibizumab (Lucentis; Genentech, South San Francisco, CA, USA) (*Rosenfeld et al., 2006*; *Brown et al., 2006*) and aflibercept (Eylea; Regeneron, Tarrytown, NY, USA, and Bayer HealthCare, Berlin, Germany)

(*Heier et al., 2012*). These anti-VEGF drugs are effective and safe in the treatment of n-AMD (*Ricci et al., 2020*; *Santarelli et al., 2015*; *Parravano et al., 2021*; *Luu et al., 2022*). According to the study on the safety and efficacy of intravitreal injection of anti-VEGF drugs, brolucizumab as a new anti-VEGF drug opens a new avenue for treating n-AMD.

Brolucizumab is a humanized single-chain antibody fragment comprising the tips of the Fab region of the antibody and linked by a peptide linker. Like ranibizumab, brolucizumab inhibits all isoforms of VEGF-A. Therefore, we conducted this meta-analysis to compare the efficacy and safety of brolucizumab *vs.* other anti-VEGF drugs in the treatment of n-AMD (*Agostini et al., 2020*; *Zhang et al., 2021*; *Haensli, Pfister & Garweg, 2021*; *Ota et al., 2022*; *Holz et al., 2022*; *Khanani et al., 2022*; *Khoramnia et al., 2022*).

Furthermore, brolucizumab (*Sharma et al., 2021*; *Bulirsch et al., 2022*) is a novel monoclonal antibody anti-VEGF drug for the treatment of n-AMD, with more research significance in its efficacy and safety than other anti-VEGF drugs (*Sharma et al., 2021*; *Tamashiro et al., 2022*; *Agostini et al., 2020*; *Bulirsch et al., 2022*).

Although several network meta-analyses have compared the efficacy of brolucizumab with other anti-VEGF drugs, no meta-analysis on the safety and efficacy of brolucizumab has been published (*Baumal et al., 2020*; *Yu et al., 2021*). Thereby, we conducted this meta-analysis to analyze the safety and efficacy of brolucizumab for n-AMD, and to provide references for the clinical treatment of n-AMD.

## MATERIALS AND METHODS

This meta-analysis was registered with the International Prospective Register of Systematic Reviews (PROSPERO) (CRD42023389716).

### Literature search

Relevant studies were retrieved from Cochrane, PubMed, Embase, and Web of Science. According to the "PICO" retrieval strategy, "Brolucizumab" and "Macular Degeneration" were used as search words. All studies were retrieved using subject headings and free words, and the reference lists of the included studies were also manually searched to avoid missing eligible studies. The search strategy () is described in Table S1. The literature retrieval was performed by Ran Dou and Jian Jiang and Dongchang Zhang served as a referee. However, there were no disagreements between the researchers.

### Inclusion and exclusion criteria

#### Inclusion criteria

1) Patients: patients were diagnosed with n-AMD;
2) Intervention: the patients in experimental groups received the intravitreal injection of Brolucizumab;
3) Comparison: the patients in the control groups received the intravitreal injection of other anti-VEGF drugs, including aflibercept or ranibizumab, which were recognized as effective drugs for AMD.
4) Outcomes: the primary outcome measures included best-corrected visual acuity (BCVA) and central subfield thickness (CSFT), while the secondary outcomes were

the presence of IRF and/or SRF, presence of sub-RPE fluid, participants with ≥1 serious adverse events, and participants with ≥1 adverse event.

5) Study type: randomized controlled trials (RCTs).

*Exclusion criteria*

1) Animal experiments, meta-analyses, meeting abstracts, letters, or systematic reviews;
2) Duplicate literature or suspected plagiarism;
3) unavailable original text;
4) unavailable data of interest;
5) Non-English literature.

## Data extraction

Literature screening was independently conducted by two researchers (Ran Dou and Jian Jiang) according to the inclusion and exclusion criteria. Then, relevant data were extracted and cross-checked. Extracted data included basic information about the selected studies (*e.g.*, title, first author, journal publication time, region), intervention measures of the control group and the experimental group, the risk of bias, and outcome measures.

## Quality evaluation

According to the Risk of Bias Assessment tool of Cochrane Collaboration, the same two investigators assessed the risk of bias in the included studies. The assessment included seven items: randomization method, allocation concealment, blinding of participants and personnel, blinding of outcome measures, incomplete outcome data, selective reporting, and other biases. Each item was evaluated as "low risk," "high risk," or "unclear.

## Statistical analysis

The STATA software was used for the meta-analysis, and the RevMan software provided by the Cochrane Collaboration was used for bias assessment.

Given the different doses used in the experimental groups, subgroup analysis was performed according to 6 mg brolucizumab and <6 mg brolucizumab. Standardized mean difference (SMD) and relative risk (RR) were used as the effect sizes, and a 95% confidence interval (CI) was provided for each effect size. Q-test was used for qualitative analysis, and the $I^2$ statistic was adopted to quantify the heterogeneity. $P > 0.1$ and $I^2 \leq 50\%$ indicated the presence of homogeneity among the studies, and a fixed-effects model was used for meta-analysis. Otherwise, no heterogeneity was determined, and a random-effects model was used for meta-analysis. In the case of clinical heterogeneity, subgroup analysis or sensitivity analysis was performed to explore the source of heterogeneity.

# RESULTS

## Literature search process and results

According to the search strategy, 511 studies were retrieved. After the duplicates were removed, 322 studies remained. After a careful review of the titles and abstracts, 290
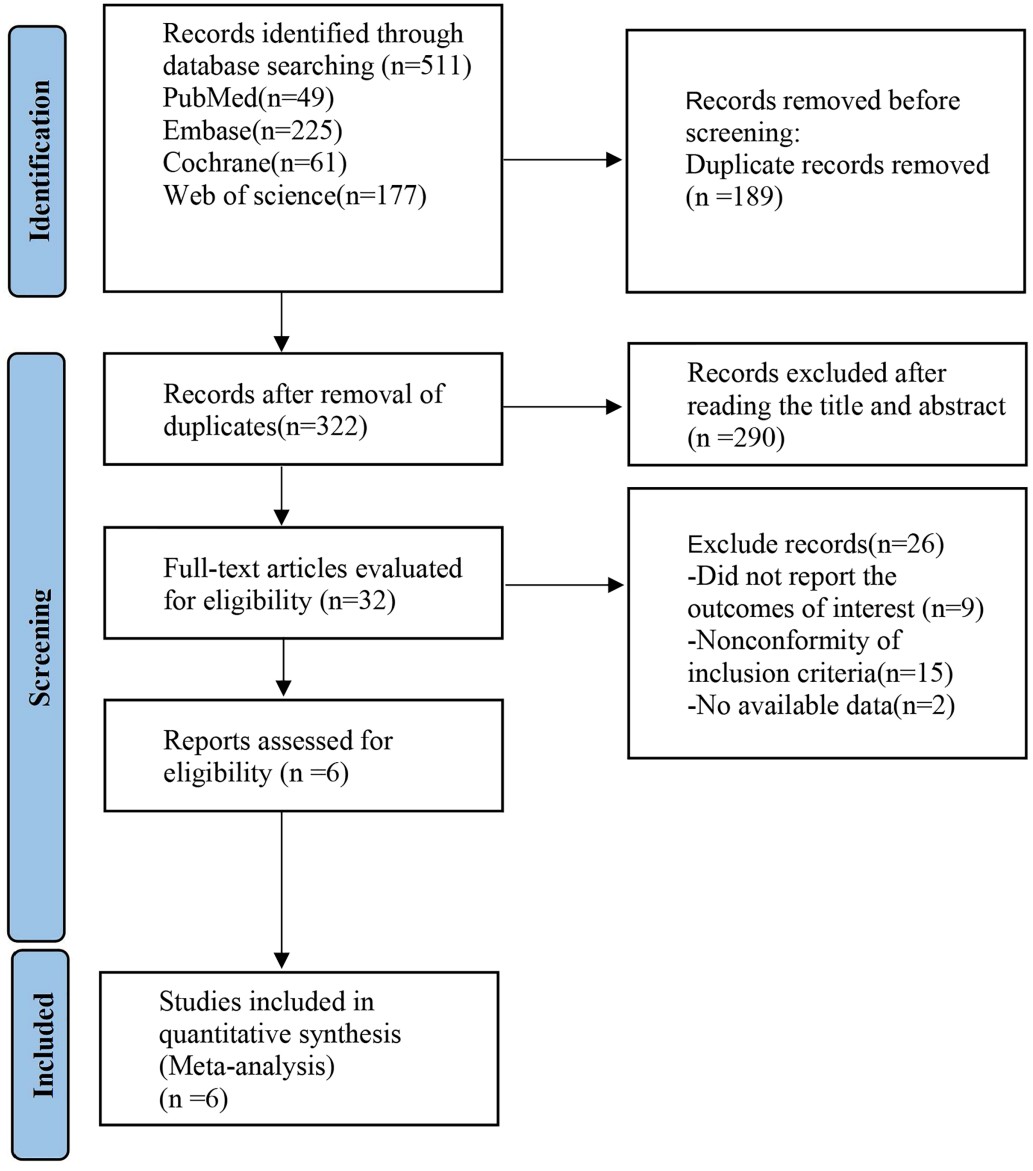

**Figure 1 Preferred Reporting Items for Systematic reviews and Meta-Analyses (PRISMA) flow diagram of the study process.**

ineligible articles were excluded. Based on further assessment of the full text, six studies were finally included. The basic characteristics of the included studies are shown in Fig. 1.

## Basic characteristics table of the included literature

The six included studies (*Agostini et al., 2020*; *Zhang et al., 2021*; *Haensli, Pfister & Garweg, 2021*; *Ota et al., 2022*; *Holz et al., 2022*; *Khanani et al., 2022*) were all RCTs. The intervention for the experimental group was an intravitreal injection of Brolucizumab, while the intervention for the control group was an intravitreal injection of other anti-VEGF drugs, such as aflibercept and ranibizumab (Table 1).
**Table 1 Basic information of the included studies.**

| Study | Experiment design | Country | Sample size (male) | | Mean age (years) | | Intervention | | Follow up time | Outcome indicators |
|---|---|---|---|---|---|---|---|---|---|---|
| | | | EG | CG | EG | CG | EG | CG | | |
| *Ogura et al. (2022)* | RCT | Japan | 39 | 30 | | | Brolucizumab 6.0 mg | Aflibercept 2.0 mg | 48 weeks | F1 F2 F3 |
| *Holz et al. (2016)* | RCT | United States, Europe, Israel, and Australia | 11 (5) | 61 (28) | 75.7 (6.5) | 77.8 (8.1) | RTH258 0.5 mg | Ranibizumb 0.5 mg | 24 weeks | F5 F6 |
| | | | 31 (19) | | 78.4 (8.3) | | RTH258 3.0 mg | | | |
| | | | 47 (21) | | 75.3 (7.7) | | RTH258 4.5 mg | | | |
| | | | 44 (15) | | 74.8 (9.8) | | RTH258 6.0 mg | | | |
| *Dugel et al. (2017)* | RCT | United States | 44 (16) | 45 (20) | 77.8 | 78.3 | Brolucizumab 6.0 mg | Aflibercept 2.0 mg | 56 weeks | F5 |
| *Dugel et al. (2020)* | RCT | Argentina, Australia, Canada, Colombia, Israel, Japan, Mexico, New Zealand, Panama, Puerto Rico, and USA. | 358 (148) | 360 (166) | 76.7 (8.28) | 76.2 (8.8) | Brolucizumab 3 mg | Aflibercept 2 mg (HAWK) | 48 weeks | F1 F2 F3 F4 F5 |
| | | | 360 (155) | | 76.7 (8.95) | | Brolucizumab 6 mg | | | |
| | | Austria, Belgium, Croatia, Czech Republic, Denmark, Estonia, Finland, France, Germany, Greece, Hungary, Ireland, Italy, South Korea, Latvia, Lithuania, Netherlands, Norway, Poland, Portugal, Russia, Singapore, Slovakia, Spain, Switzerland, Taiwan, Turkey, UK, and Vietnam | 370 (160) | 369 (157) | 74.8 (8.58) | 75.5 (7.87) | Brolucizumab 6 mg | Aflibercept 2 mg (HARRIER) | 48 weeks | F1 F2 F3 F5 F6 |
| NCT03386474 (*ClinicalTrials. gov, 2013*) | nRCT | United States | 107 (38) | 43 (21) | 80.6 (8.63) | 77.9 (9.20) | Brolucizumab 6 mg | Aflibercept 2 mg | 24 weeks | F5 |
| NCT01849692 (*ClinicalTrials. gov, 2019*) | RCT | US, Australia, the Dominican Republic | 10 (7) | 3 (2) | 77.5 (3.6) | 78.3 (7.2) | ESBA1008 1.2 mg | Ranibizumb 0.5 mg (stage1) | 56 days | F1 F2 |
| | | | 10 (4) | 3 (1) | 73.6 (8.9) | 82.0 (3.6) | ESBA1008 1.0 mg | Ranibizumb 0.5 mg (stage1) | | F1 F2 F5 |
| | | | 10 (5) | 3 (1) | 76.5 (9.7) | 81.7 (11.0) | ESBA1008 0.6 mg | Ranibizumb 0.5 mg (stage2) | | F1 F2 F5 |
| | | | 10 (6) | 3 (2) | 81.6 (6.1) | 76.7 (3.8) | ESBA1008 0.5 mg | Ranibizumb 0.5 mg (stage2) | | F1 F2 F5 |

Notes:
RCT, randomized controlled trial; EG, the experimental group CG: the control group.
F1, BCVA(best-corrected visual acuity); F2, CSFT(central subfield thickness); F3, Presence of IRF and/or SRF, $n$ (%); F4, Presence of sub-RPE fluid, $n$ (%); F5, safety1, Participants with ≥1 serious adverse event, $n$ (%); F6, safety2, Participants with ≥1 adverse event, $n$ (%).
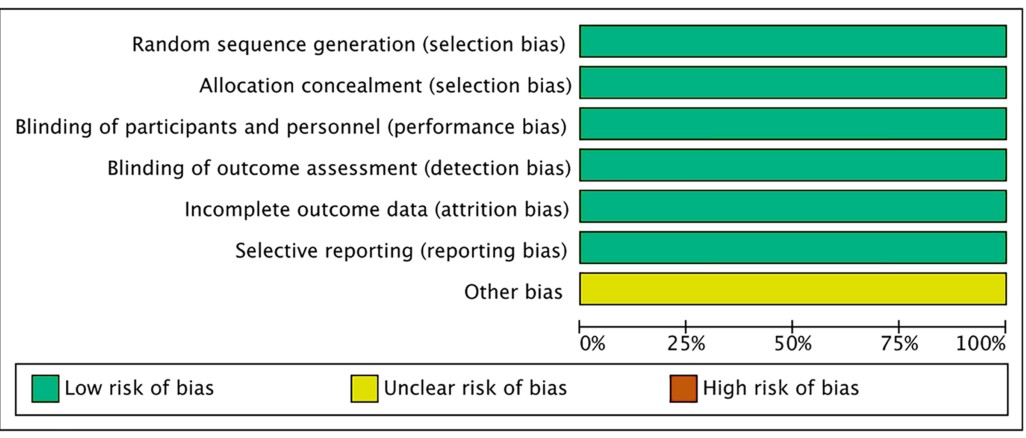

**Figure 2** Risk of bias graph of all the retrieved studies.

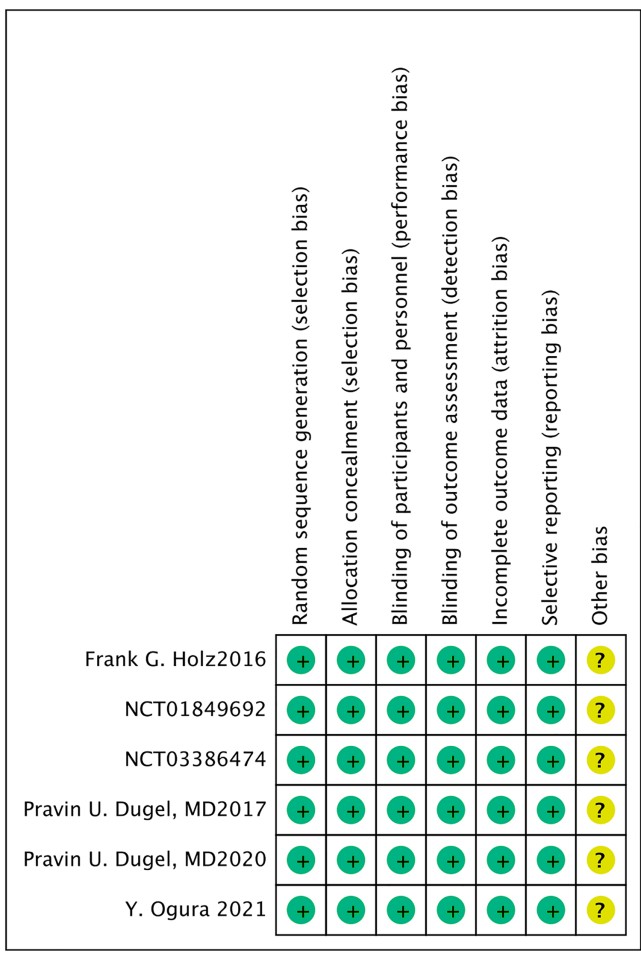

**Figure 3** Risk of bias summary of all the retrieved studies.

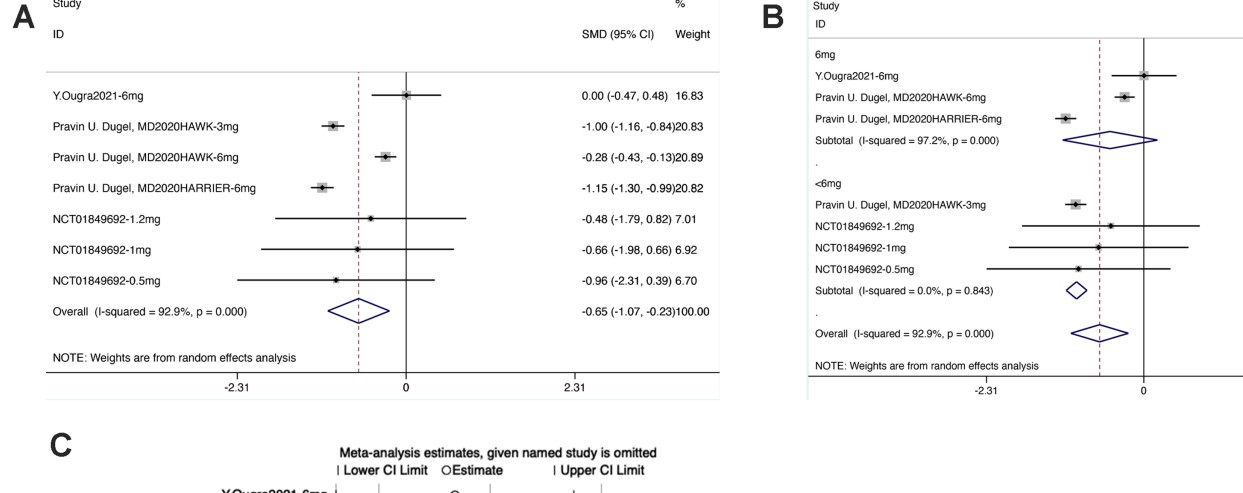

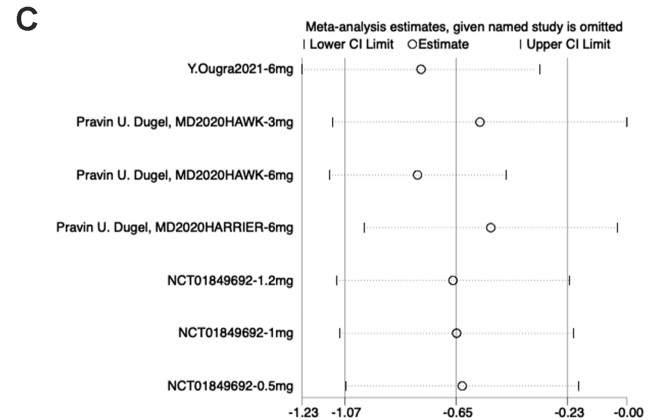

**Figure 4 (A–C) Meta-analysis for BCVA.**

## Risk of bias assessment

All six included studies (*Agostini et al., 2020*; *Zhang et al., 2021*; *Haensli, Pfister & Garweg, 2021*; *Ota et al., 2022*; *Holz et al., 2022*; *Khanani et al., 2022*) were randomized, double-blind, multicenter clinical trials. All the included studies mentioned the allocation concealment, the blinding of participants and personnel, as well as the blinding of outcome measurement (Figs. 2, 3).

## Meta-analysis results

### BCVA

From the forest plot, the diamond was on the left of the invalid line and did not intersect with the invalid line in the Brolucizumab group. The results showed that the effect of intravitreal injection of Brolucizumab on BCVA was inferior to that of the control intervention [SMD = −0.65, 95% CI [−0.17 to −0.23], $P < 0.05$], with statistically significant differences (Fig. 4).

As shown in the forest plot for subgroup analysis, the diamond was on the left of the invalid line and intersected with the invalid line in the experimental subgroup of 6 mg Brolucizumab. No significant difference was noted between the subgroup of 6 mg Brolucizumab and the control group [SMD = −0.50, 95% CI [−1.19 to −0.20], $P > 0.05$]. In the experimental subgroup of <6 mg Brolucizumab, the diamond was on the right of the

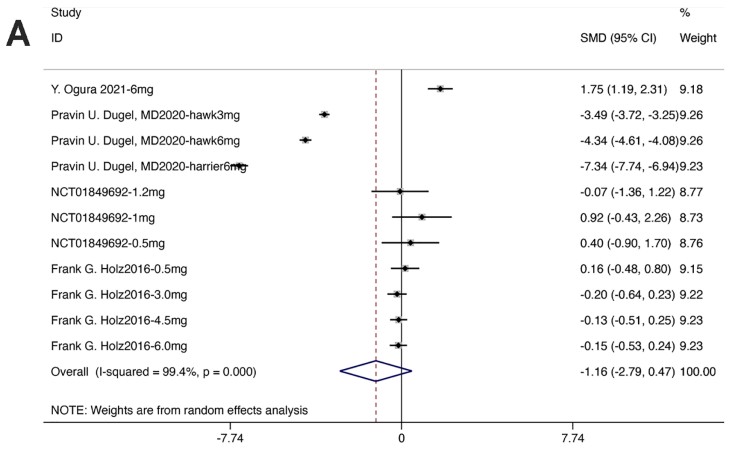

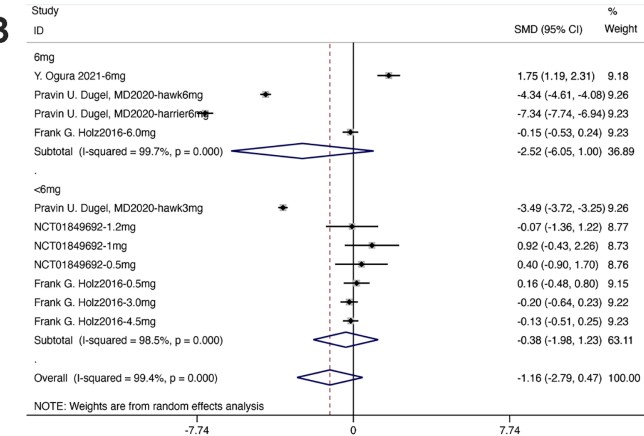

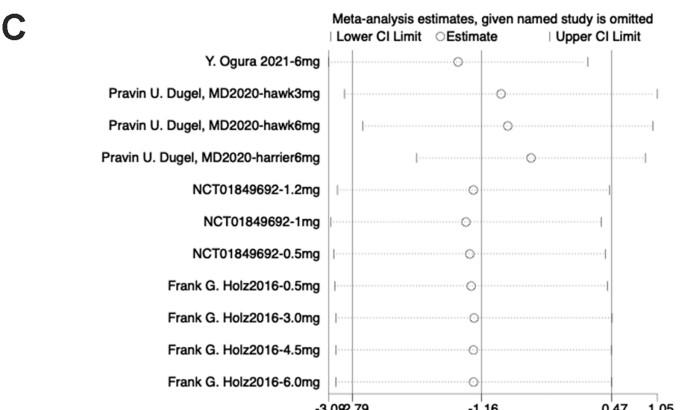

**Figure 5 (A–C) Meta-analysis for CSFT.**

invalid line and did not intersect with the invalid line, indicating significant differences between the experimental subgroup of <6 mg Brolucizumab and the control group [SMD = −099, 95% CI [−1.14 to −0.84], $P$ < 0.05].

Sensitivity analysis revealed that the data were distributed within the interval, indicating stable results.

### CSFT

The forest plot exhibited that the diamond intersected with the invalid line in the experimental group, indicating no statistical difference between the experimental group and the control group [SMD = −1.16, 95% CI [−2.79 to 0.47], $P$ > 0.05] (Fig. 5).

From the forest plot for subgroup analysis, in the experimental subgroups of both 6 mg brolucizumab and <6 mg brolucizumab, the diamond intersected with the invalid line, indicating no statistical difference between the experimental subgroup and the control group [6 mg Brolucizumab: SMD = −2.52, 95% CI [−6.05 to 1.00], $P$ > 0.05; <6 mg Brolucizumab: SMD = −0.38, 95% CI [−1.98 to 1.23], $P$ > 0.05].

Sensitivity analysis manifested that the data were distributed within the interval, indicating stable results.

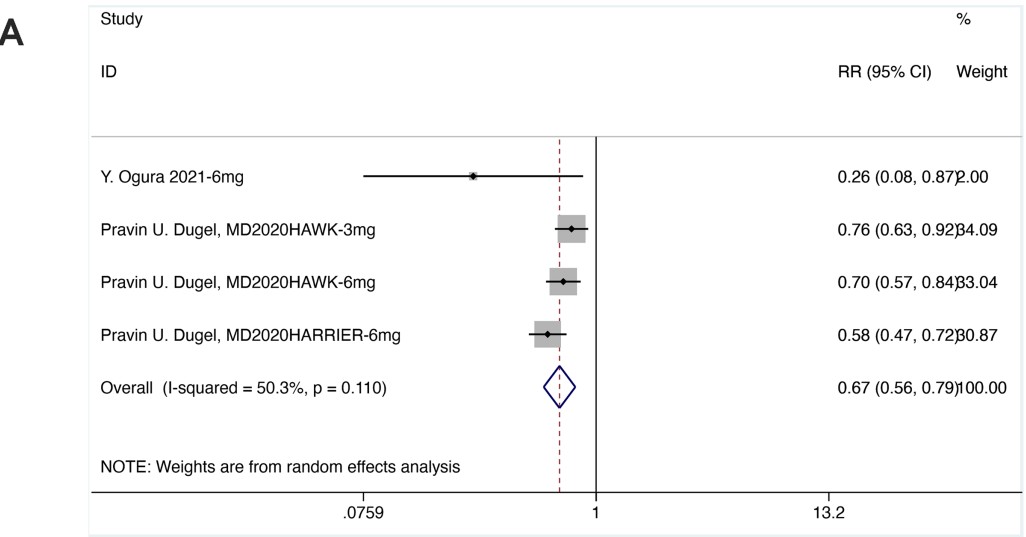

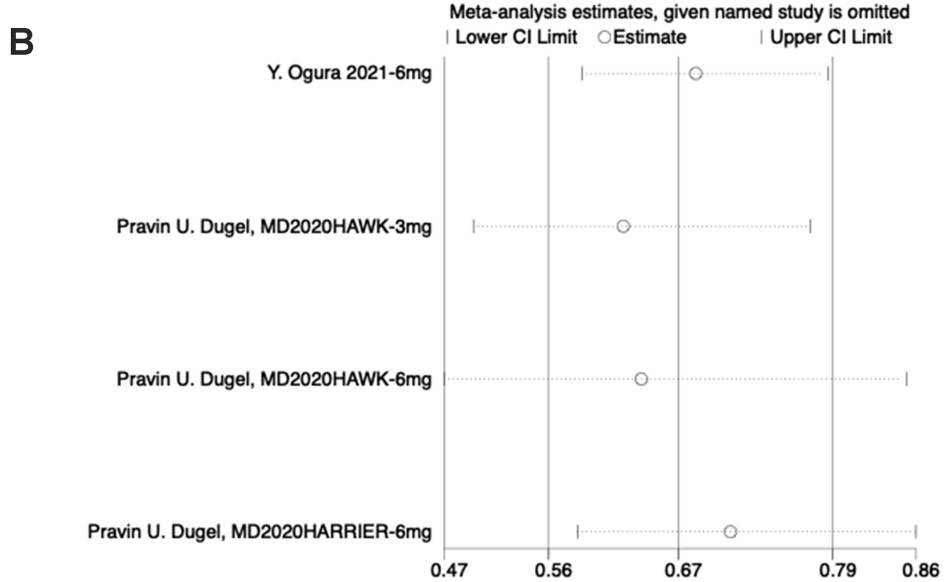

**Figure 6** (A and B) Meta-analysis for presence of IRF and/or SRF (F3).

## Presence of IRF and/or SRF (F3)

As shown in the forest plot, in the experimental group, the diamond was on the left of the invalid line and did not intersect with the invalid line, suggesting statistical differences between the experimental group and the control group [RR = 0.67, 95% CI [0.56–0.79], $P < 0.05$].

Sensitivity analysis noted that the data were distributed within the interval, indicating stable results (Fig. 6).

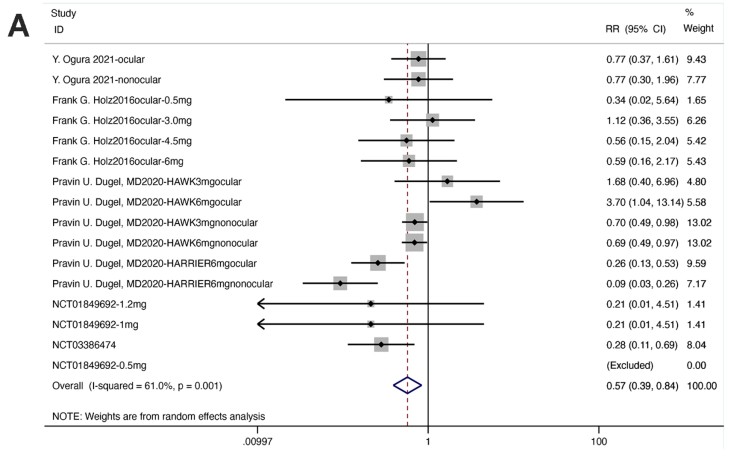

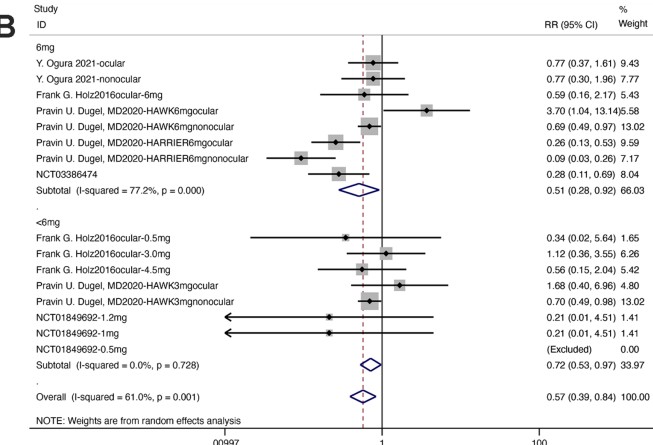

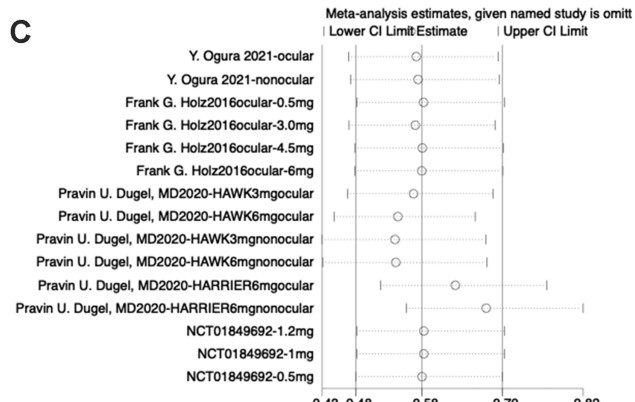

**Figure 7** (A–C) Sensitivity analysis.

### Participants with ≥1 serious adverse events (F5)

From the forest plot, in the experimental group, the diamond intersected with the invalid line, indicating statistical differences between the experimental group and the control group [RR = 0.57, 95% CI [0.39–0.84], $P < 0.05$].

From the forest plot for subgroup analysis, in the experimental subgroup of 6 mg brolucizumab, the diamond was on the left of the invalid line and did not intersect with the invalid line, demonstrating statistical differences between the experimental subgroup of 6 mg Brolucizumab and the control group [RR = 0.51, 95% CI [0.28–0.92], $P < 0.05$]. In the subgroup of <6 mg brolucizumab, the diamond was on the left of the invalid line and intersected with the invalid line, suggesting no statistical differences between the experimental subgroup of <6 mg brolucizumab and the control group [RR = 0.72, 95% CI [0.53–0.97], $P > 0.05$].

Sensitivity analysis manifested that the data were distributed within the interval, indicating stable results (Fig. 7).

### Participants with ≥1 adverse events (F6)

As shown in the forest plot, in the experimental group, the diamond was on the right of the invalid line and intersected with the invalid line, demonstrating no statistical difference

between the experimental group and the control group [RR = 1.07, 95% CI [0.97–1.17], $P > 0.05$].

From the forest plot for subgroup analysis, in the experimental subgroup, the diamond was on the right of the invalid line and intersected with the invalid line, demonstrating no statistical difference between the experimental subgroup and the control group [6 mg brolucizumab: RR = 1.06, 95% CI [0.91–1.23], $P > 0.05$; <6 mg brolucizumab: RR = 1.07, 95% CI [0.95–1.26], $P > 0.05$].

Sensitivity analysis revealed that the data were distributed within the interval, indicating stable results (Fig. S1).

## Evaluation of publication bias

The outcome measures of this study covered BCVA, CSFT, presence of IRF and/or SRF, participants with ≥1 serious adverse events, and participants with ≥1 adverse events. The funnel plots drawn for these outcome measures visually observed asymmetric distribution on both sides, indicating the presence of publication bias (Fig. S2).

## DISCUSSION

In this study, six studies (*Yu et al., 2021*; *Dugel et al., 2017*; *ClinicalTrials.gov, 2013*, *2019*; *Holz et al., 2016*; *Dugel et al., 2020*) were included for meta-analysis to evaluate the efficacy and safety of different anti-VEGF drugs on n-AMD based on BCVA, CSFT, presence of IRF and/or SRF, participants with ≥1 serious adverse events, and participants with ≥1 adverse events (*Tamashiro et al., 2022*; *Zakaria et al., 2022*; *Montesel et al., 2021*). Subgroup analysis was performed according to different doses of Brolucizumab.

In clinical medication, the safety of brolucizumab is of utmost priority. Three studies (*Yu et al., 2021*; *ClinicalTrials.gov, 2019*; *Holz et al., 2016*) reported the safety of brolucizumab in participants with ≥1 adverse events in n-AMD. Our meta-analysis showed no statistical difference between the experimental group and the control group, indicating that intravitreal injection of brolucizumab is equally safe compared to other anti-VEGF drugs in participants with ≥1 adverse events. In addition, the meta-analysis results of five studies (*ClinicalTrials.gov, 2013*, *2019*; *Holz et al., 2016*; *Dugel et al., 2020*; *Ogura et al., 2022*) showed that intravitreal injection of 6 mg brolucizumab yielded better results regarding the safety (*Angermann et al., 2022*) of participants with ≥1 serious adverse events. In summary, brolucizumab is as safe as other anti-VEGF drugs for n-AMD patients, especially for those with serious adverse events, which provides references for its safety in clinical application.

Given the equal safety to other anti-VEGF drugs, efficacy has become the most important indicator of clinical medication. BCVA, CSFT, and the presence of IRF and/or SRF are recognized aspects to determine the efficacy of Brolucizumab in the treatment of n-AMD. The meta-analysis results of three studies (*Dugel et al., 2017*; *Holz et al., 2016*; *Dugel et al., 2020*) on BCVA showed that intravitreal injection of 6 mg brolucizumab was as effective as the control, although other concentrations of brolucizumab were less

effective. Therefore, Brolucizumab at a dose of 6 mg can yield similar outcomes on BCVA as aflibercept. As for CSFT, the meta-analysis results of previous research showed no advantage for brolucizumab (*Dugel et al., 2022*; *Boltz et al., 2022*). Our present study also found that brolucizumab did not outperform the control drugs in CSFT. Would >6 mg brolucizumab lead to better results on BCVA and CSFT based on non-inferior safety to other anti-VEGF drugs? Which dose would be the best? These issues should be further addressed in the future.

The meta-analysis results noted that intravitreal injection of brolucizumab outperformed aflibercept in the presence of IRF and/or SRF. In treatment cases, we found that the brolucizumab group showed better results than the control group on morphology. Furthermore, in terms of molecule characteristics, like ranibizumab, brolucizumab specifically inhibits VEGF-A. However, due to its low molecular weight and high stability and solubility, it can be highly concentrated and administered in a molar dose 12 times that of aflibercept and 22 times that of ranibizumab (*Bulirsch et al., 2022*). The anatomical impact of such high-dose brolucizumab on the choroid might be more successful in addressing SRF and IRF. Combined with the above-mentioned effects on BCVA and CSFT, although brolucizumab can be accumulated in high concentrations, it did not show an advantage in reducing CSFT, despite its excellent morphological performance. In brief, if brolucizumab could have therapeutic effects on CSFT, its efficacy in n-AMD may be more significant.

In the global phase III HAWK and HARRIER trials, intravitreal injections of brolucizumab every 3 months after the loading phase yielded similar visual outcomes to intravitreal injections of aflibercept every 2 months (*Bulirsch et al., 2022*), which is a more convenient and cheaper choice for n-AMD patients. Brolucizumab might have better therapeutic effects for clinical treatment. However, the evaluation indicators of the included studies varied, and the sample size of the same outcome indicators was limited. Therefore, the results of this meta-analysis should be interpreted with caution. Additionally, the literature search was limited to English papers, and some high-quality studies reported in other languages may have been missed. Multicenter and high-quality clinical studies with larger samples are needed to validate the efficacy and safety of brolucizumab.

## CONCLUSIONS

Compared to other anti-VEGF drugs such as aflibercept and ranibizumab, intravitreal injection of 6 mg brolucizumab is more effective and safer in the treatment of n-AMD, especially in the presence of IRF and/or SRF, and on participants with ≥1 serious adverse events.

### Funding
The authors received no funding for this work.

## Competing Interests

The authors declare that they have no competing interests.

## Author Contributions

- Ran Dou conceived and designed the experiments, performed the experiments, analyzed the data, prepared figures and/or tables, authored or reviewed drafts of the article, and approved the final draft.
- Jian Jiang conceived and designed the experiments, authored or reviewed drafts of the article, and approved the final draft.

## Data Availability

This is a systematic review/meta-analysis.

## Supplemental Information

Supplemental information for this article can be found online at http://dx.doi.org/10.7717/peerj.17561#supplemental-information.

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

# PeerJ

**Montesel A, Bucolo C, Sallo FB, Eandi CM. 2021.** Short-term efficacy and safety outcomes of brolucizumab in the real-life clinical practice. *Frontiers in Pharmacology* **12**:720345 DOI 10.3389/fphar.2021.720345.

**Ogura Y, Jaffe GJ, Cheung CMG, Kokame GT, Iida T, Takahashi K, Lee WK, Chang AA, Monés J, D'Souza D, Weissgerber G, Gedif K, Koh A. 2022.** Efficacy and safety of brolucizumab versus aflibercept in eyes with polypoidal choroidal vasculopathy in Japanese participants of HAWK. *British Journal of Ophthalmology* **106(7)**:994–999 DOI 10.1136/bjophthalmol-2021-319090.

**Ota H, Takeuchi J, Nakano Y, Horiguchi E, Taki Y, Ito Y, Terasaki H, Nishiguchi KM, Kataoka K. 2022.** Switching from aflibercept to brolucizumab for the treatment of refractory neovascular age-related macular degeneration. *Japanese Journal of Ophthalmology* **66(3)**:278–284 DOI 10.1007/s10384-022-00908-1.

**Parravano M, Costanzo E, Scondotto G, Trifirò G, Virgili G. 2021.** Anti-VEGF and other novel therapies for neovascular age-related macular degeneration: an update. *BioDrugs* **35(6)**:673–692 DOI 10.1007/s40259-021-00499-2.

**Ricci F, Bandello F, Navarra P, Staurenghi G, Stumpp M, Zarbin M. 2020.** Neovascular age-related macular degeneration: therapeutic management and new-upcoming approaches. *International Journal of Molecular Sciences* **21(21)**:8242 DOI 10.3390/ijms21218242.

**Rosenfeld PJ, Brown DM, Heier JS, Boyer DS, Kaiser PK, Chung CY, Kim RY. 2006.** Ranibizumab for neovascular age-related macular degeneration. *New England Journal of Medicine* **355(14)**:1419–1431 DOI 10.1056/NEJMoa054481.

**Santarelli M, Diplotti L, Samassa F, Veritti D, Kuppermann BD, Lanzetta P. 2015.** Advances in pharmacotherapy for wet age-related macular degeneration. *Expert Opinion on Pharmacotherapy* **16(12)**:1769–1781 DOI 10.1517/14656566.2015.1067679.

**Sharma A, Kumar N, Parachuri N, Sadda SR, Corradetti G, Heier J, Chin AT, Boyer D, Dayani P, Arepalli S, Kaiser P. 2021.** Brolucizumab-early real-world experience: BREW study. *Eye (Lond)* **35(4)**:1045–1047 DOI 10.1038/s41433-020-1111-x.

**Takahashi K, Iida T, Ishida S, Crawford B, Sakai Y, Mochizuki A, Tsujiuchi R, Tanaka S, Imai K. 2022.** Effectiveness of current treatments for wet age-related macular degeneration in japan: a systematic review and pooled data analysis. *Clinical Ophthalmology* **16**:531–540 DOI 10.2147/OPTH.S345403.

**Tamashiro T, Tanaka K, Itagaki K, Nakayama M, Maruko I, Wakugawa S, Terao N, Onoe H, Wakatsuki Y, Ogasawara M, Sugano Y, Yamamoto A, Kataoka K, Izumi T, Kawai M, Mori R, Sekiryu T, Okada AA, Iida T, Koizumi H. 2022.** Subfoveal choroidal thickness after brolucizumab therapy for neovascular age-related macular degeneration: a short-term multicenter study. *Graefes Archive for Clinical and Experimental Ophthalmology* **260(6)**:1857–1865 DOI 10.1007/s00417-021-05517-1.

**Wong TY, Chakravarthy U, Klein R, Mitchell P, Zlateva G, Buggage R, Fahrbach K, Probst C, Sledge I. 2008.** The natural history and prognosis of neovascular age-related macular degeneration: a systematic review of the literature and meta-analysis. *Ophthalmology* **115(1)**:116–126 DOI 10.1016/j.ophtha.2007.03.008.

**Yu JS, Carlton R, Agashivala N, Hassan T, Wykoff CC. 2021.** Brolucizumab vs aflibercept and ranibizumab for neovascular age-related macular degeneration: a cost-effectiveness analysis. *Journal of Managed Care & Specialty Pharmacy* **27(6)**:743–752 DOI 10.18553/jmcp.2021.27.6.743.

**Zakaria N, Guerard N, Emanuelli A, Dugel P, Watts J, Liew M, Gekkieva M, Hinder M. 2022.** Evaluation of cardiac parameters and other safety outcomes of brolucizumab treatment in

patients with neovascular age-related macular degeneration. *Pharmacology Research & Perspectives* **10(2)**:e00897 DOI 10.1002/prp2.897.

**Zhang Y, Gao S, Li X, Huang X, Zhang Y, Chang T, Cai Z, Zhang M. 2021.** Efficacy and safety of anti-vascular endothelial growth factor monotherapies for neovascular age-related macular degeneration: a mixed treatment comparison. *Frontiers in Pharmacology* **12**:797108 DOI 10.3389/fphar.2021.797108.