# Peer review of "Efficacy and safety of Brolucizumab for neovascular age-related macular degeneration: a systematic review and meta-analysis"

_PeerJ, doi:10.7717/peerj.17561_

## Round 0.1 · original submission · Major Revisions

Before this article is published, the authors need to address the comments of the reviewers.

**Language Note:** The review process has identified that the English language must be improved. PeerJ can provide language editing services - please contact us at copyediting@peerj.com for pricing (be sure to provide your manuscript number and title). Alternatively, you should make your own arrangements to improve the language quality and provide details in your response letter. – PeerJ Staff

Reviewer 2 ·

Basic reporting

The study under review conducts a thorough examination of the effects and safety of various anti-VEGF drugs in treating neovascular age-related macular degeneration. While the research presents intriguing findings, it has several areas that require improvement. The reviewer emphasizes the need for clarity in statistical reporting, proper presentation of abbreviations, adherence to reference formatting standards, enhancement of figure legibility, and significant language editing for coherence and readability. The research is interesting, but there are some weaknesses needed to be addressed as detailed belows.

1.The manuscript presents a comprehensive analysis, including statistical differences in BCVA, presence of IRF/SRF, and safety regarding serious adverse events. However, for a more robust scientific presentation, it is crucial to supplement the P values for these findings. This addition will not only enhance the statistical robustness of the results but also provide clarity on the significance levels of the findings.
2.The abstract contains several abbreviations like BCVA (Best-Corrected Visual Acuity), CSFT (Central Subfield Thickness), IRF (Intraretinal Fluid), and SRF (Subretinal Fluid). While these terms are common in ophthalmological research, spelling them out in the abstract will enhance readability and accessibility for a broader audience, including those not specialized in the field.
3.The manuscript currently lacks proper citation and presentation of references as per formatting guidelines. This can lead to challenges in verifying and consulting the source material. Adherence to a consistent reference format is essential for the academic integrity and credibility of the research.
4.The figures included in the study, such as the forest plots for BCVA and CSFT, are integral to understanding the results. However, their impact is diminished by small font sizes and clarity issues. Improving these aspects will significantly enhance the reader's ability to interpret and analyze the data presented.
5.While the research content is substantial, the language quality requires rigorous editing. This includes refining the grammar, syntax, and overall fluency to meet the standards expected in scientific publications. Enhanced language quality will not only improve readability but also ensure that the research is accurately communicated to the global scientific community.
6.Additionally, it would be beneficial to address any potential publication bias, as indicated by the asymmetrical scatter distribution in the funnel plots for the outcome measures. Ensuring a balanced representation of studies and discussing any limitations in this regard will add to the manuscript's transparency and reliability.

Overall, these enhancements will significantly improve the manuscript's quality, making the valuable findings on the efficacy and safety of Brolucizumab and other anti-VEGF drugs in treating n-AMD more impactful and accessible to the scientific community.

Experimental design

none

Validity of the findings

none

Additional comments

none

·

Basic reporting

The authors of this article found that intravitreal injection of 6 mg Brolucizumab for neovascular AMD is a relatively effective and safe treatment for n-AMD, which provides some assistance in the clinical management of n-AMD. Some modifications are needed.
All abbreviations in the manuscript should be marked with their full names for the first time.
It is suggested to supplement the Reference search time.
Authors should refer to the published literature and express the 95%CI value correctly.
The Introduction should be further refined.
The manuscript was poorly written. A large number of typos and grammatical errors were seen throughout the whole manuscript.
In the discussion section, the authors need to emphasize and elaborate on the novelty aspect of their work. Also, they need to expand on the clinical applicability of their findings.
Please improve the figure font and clarity.

Experimental design

See Basic reporting

Validity of the findings

See Basic reporting

Additional comments

See Basic reporting

---

## Round 0.2 · accepted · Accept

The authors have revised the manuscript well. The manuscript can now be accepted.

Reviewer 2 ·

Basic reporting

The authors have revised the manuscript well. The manuscript can now be accepted.

Experimental design

The manuscript can now be accepted.

Validity of the findings

The manuscript can now be accepted.

Additional comments

The manuscript can now be accepted.

·

Basic reporting

No more comments.

Experimental design

No more comments.

Validity of the findings

No more comments.

Additional comments

No more comments.